# Microorganisms in Fish Feeds, Technological Innovations, and Key Strategies for Sustainable Aquaculture

**DOI:** 10.3390/microorganisms11020439

**Published:** 2023-02-09

**Authors:** Pallab K. Sarker

**Affiliations:** Environmental Studies Department, University of California Santa Cruz, Santa Cruz, CA 95064, USA; psarker@ucsc.edu

**Keywords:** microorganisms, microalgae, yeast, bacteria, fungi, fishmeal, fish oil, aquaculture, environment, sustainability, feed, digestibility, nutrients, lipid, fatty acid, omega-3

## Abstract

Aquaculture, the world’s fastest growing food sector, produces over half of all fish for human consumption. Aquaculture feeds include fishmeal and fish oil, extracted from wild-caught fish such as sardines, and poses ecological, food security, and economic drawbacks. Microalgae, yeasts, fungi, bacteria, and other alternative ingredients show promise as potential ingredients in aquafeeds that provide protein/amino acids, lipids, or omega-3 sources and sources of bioactive molecules. This review article discusses the issues that the literature often lacks data on, such as the recent development of using microorganisms, technological innovation, challenges, and opportunities to develop a low environmental footprint of aquaculture diet. The ingredients often require novel processing technology to improve digestibility and fish growth and reduce antinutritional factors. This is an important gap to fill because microalgae are the most frequently used organism in fish feed, particularly as a dietary supplement or mixed with other ingredients. The production, processing, and formulating steps can affect the nutritional qualities. Stepwise strategies are required to evaluate these ingredients for feed application, and in this article, I articulated the stepwise key approaches of evaluating nutritional and environmental response metrics to develop highly sustainable aquaculture feed using these microorganisms, which would guide a more judicious inclusion of these novel ingredients.

## 1. Significance of Aquaculture and Aquafeed

Aquaculture, the fastest growing food sector globally (8% average production increase/yr. for 1970–2014), now produces about half of all fish for human consumption, as global capture fisheries have reached or exceeded their sustainable limits and plateaued at ~90 million tons/yr. [1,2,3,4]. Aquaculture is the world’s most efficient protein generator and will play a key role in solving the grand challenge of feeding more than 9 billion people by 2050 by meeting the escalating demand for water, energy, and food while conserving environmental resources and protecting livelihoods [5,6,7,8,9,10,11]. The U.S.’s demand for seafood has contributed to global aquaculture’s spectacular rise. The demand for fish and seafood has constantly increased, and aquaculture has increasingly filled this demand for human consumption, making it a fundamental part of future food production. Aquaculture is expected to continue its remarkable rise and is projected to produce 109 million metric tons of fish in 2030 for global consumers. Fed aquaculture grew 158% from 2000 to 2018 when it comprised nearly 60 million MT of production [3]. The production of aquaculture feeds for the fed aquaculture is likewise expected to increase, and 73.15 million tons of feeds are projected to be used by 2025 [12].

The environmental impacts and increasing price of aquaculture feed (aquafeeds) constrain U.S. aquaculture development [13]. Feed has the largest overall environmental impact along the supply chain of intensive fed aquaculture, wherein the yields depend on giving animals nutritionally complete formulated feeds [12]. The environmental sustainability of aquaculture food production systems is of critical concern due to its rapid expansion. Among the parameters that contribute to the overall environmental impacts, aquafeed was identified as an impact hotspot, and the production of feed ingredients contributed most to the global warming potential and other emissions [14]. The production, processing, and supply chain of feed ingredients are the major contributors of the greenhouse gas (GHG) footprint of aquaculture [15,16,17]. The use of terrestrial crops and the associated land-use change, the harvesting of wild-caught fish (e.g., anchovy, sardine, etc.) for fishmeal and fish oil, and the associated processing and transportation occur often and are commonly attributed to the greenhouse gas emissions in aquaculture [17,18,19,20,21,22]. As aquaculture is the fastest growing food production sector, we need to urgently address the greater GHG footprints associated with the production, processing, and complex supply chain of the aquafeed ingredients [3,23]. The environmental GHG emissions from aquaculture can by decreased by altering the composition of the feed and reducing eutrophication [24,25,26]. GHG emissions are also linked with the feed conversion ratio, FCR (the amount of feed required for each kilogram of fish produced), and improvements in the FCR (via lowering the FCR) can be achieved through innovation in feed composition [11,17].

High-performing novel ingredients and the associated production of microalgae feed can lead to an improved protein efficiency ratio, PER (the amount of protein required for each kilogram of fish protein produced); digestibility (how well fish can digest the nutrient in the ingredients); feed waste to the environment; and GHG emissions [14,17,27,28,29]. Aquafeed ingredients represent 40–75% of the total cost and 75–85% of the variable costs of aquaculture production, and these are key market drivers for aquaculture production [30,31]. The aquafeed market is also growing and expected to grow 8–10% per year, reaching $185 billion by 2023 [1,32]. The production of industrial aquaculture feeds is likewise expected to increase; 73.15 million tons of compound feeds are projected to be used by 2025 [12]. Approximately 51.3 million tons were produced in 2017.

### 1.1. Use of Fishmeal and Fish Oil in Aquafeed

Unfortunately, the fishmeal protein and fish oil ingredients in current commercial aquaculture feed (aquafeed) come from unsustainably sourced marine forage fish that are core components of marine food webs. Currently, aquafeeds use more than 70% of the world’s fishmeal and fish oil, which are rendered from small wild-caught forage fish (such as herrings, sardines, and anchovies). Globally, approximately 16.9 million of the 29 million tons of ocean-caught small fish are currently used for aquaculture feeds each year [33]. Farmed salmonids (Atlantic salmon and rainbow trout) are the largest users of fishmeal and fish oil (FMFO), and globally, these farmed species used approximately 24% of the total FM and 50% of the total FO in 2010 [34]. Higher-trophic-level marine finfish and freshwater crustaceans are the largest users of FMFO in aquafeeds, but low-trophic-level finfish species such as carp, tilapia, catfish, and milkfish are also fed with 2% to 4% of FMFO in their feeds [35,36].

Even with the diminishing inclusion of fishmeal in aquaculture feeds, an estimated shortage ranging from 0.4 to 1.32 million metric tons of fishmeal could occur by 2050, significantly impairing the aquaculture industry growth [37]. Analysts also projected that at the current rates of FM and FO consumption, aquaculture feed demands could outstrip the supply of forage fish by 2037, with disastrous consequences for the food security of billions of humans and for wild marine fish, mammals, and seabirds that forage on them [10]. Moreover, aquacultures face increased competition for FMFO from companies producing food supplements, pharmaceuticals, and feeds for other animals [35,38], causing a supply–demand squeeze: FMFO prices will keep rising steeply and supplies will be insufficient to meet growing aquafeed demands and will thus constrain aquaculture growth [1,39].

### 1.2. Use of Terrestrial Crops and Oils in Aquafeeds and Environmental Challenges

Aquafeed manufacturers currently over-rely on terrestrial crops (e.g., soy, corn, canola) to replace fishmeal and fish oil; these would otherwise feed livestock or people [10,40,41]. Industrial agriculture is facing a tremendous challenge due to agricultural pollution resulting in a loss of arable land. The increasing demand for these crops by aquaculture’s explosive growth could elevate the environmental problems that are caused by farming them, including high nutrient and chemical inputs, runoff that increases eutrophication, the clearing of sensitive lands (e.g., in Amazonia), high energy inputs, and greenhouse gas emissions [38,42,43]. However, typically, the cultivation of terrestrial crops such as transgenic soy requires a large volume of water, pesticides, and fertilizers, and these also cause the deforestation of areas with high biological value. The monoculture practice of growing soy destroys biodiversity, pollutes the soil, and depletes water resources [44,45,46]. The major soy-producing countries (USA, Brazil, and Argentina) account for 80% of global production. Soy is an international commodity, and huge exports further increase the carbon and environmental footprint of soy production [45,47,48].

Current aquaculture practices only exacerbate these challenges. Developing an economic and sustainably sourced aquafeed promises great relief of the pressure on foraging fish, the increase in needed farmland, the minimization of nutrient pollution, and the catalysis of a truly sustainable aquaculture. Thus, replacing FMFO ingredients with terrestrial crop ingredients will not meet the goal of having a zero-carbon footprint and could further exacerbate the deforestation and global warming potential. To reduce the carbon footprint of aquaculture feed, we should emphasize locally sourced feed ingredients and byproduct meals, which should not be transported across the world.

#### 1.2.1. Nutritional Disadvantages of Terrestrial Crop Protein

Terrestrial plant protein and oil ingredients have several nutritional disadvantages such as a low nutrient digestibility due to high levels of antinutritional components, and a deficiency in limiting essential amino acids, for example, lysine, methionine, threonine, and tryptophan [49,50]. For example, methionine is deficient in soybean meal and lysine is deficient in cornmeal, and these are the major ingredients in current commercial aquafeeds. Replacing FM protein with terrestrial crop protein remains a significant challenge for the industry and has stimulated compensatory research such as adding single amino acids to provide missing essential amino acids. Single amino acids, compared to amino acids bound in intact protein, have the following drawbacks in aquafeed: fish cannot efficiently utilize the synthetic amino acids and excrete more N metabolic waste to the environment, and the costs and life cycle environmental impacts increase [51,52,53,54]. Other common antinutritional factors include trypsin inhibitors, haemagglutinins, phytic acid, gossypol, phytoestrogens, glucosinolates, erucic acid, alkaloids, and thiaminase [55]. However, some antinutrient components can be destroyed via a heating and drying process [56]. Another unsustainable consequence of using plant ingredients in aquafeeds is that about 70% of the phosphorus in these ingredients are phytate bound, which is a very complex form. In fish that lack sufficient endogenous phytase enzymes to make bioavailable and undigested phytate, P ends up being excreted into the environment and may cause eutrophication potential [8,55,57]. Phytate also interacts with proteins, which may affect protein digestibility in trout and increase N waste outputs [58]. Phytate P in soy and other terrestrial crops cannot be digested by fish [56,59,60]. Several studies reported that the processing of ingredients or supplementations of phytase in diets improves the digestibility of P, minerals, and protein in salmonids [54,61,62]. For example, antinutritional factors can be overcome by using soybean protein concentrates and canola protein concentrate (60–65% protein), and antinutritional factors can be reduce via other processing methods [63]. The phytase enzyme can enhance the digestibility of terrestrial plant protein; however, the instability of the enzyme during feed processing is an issue. Additionally, the possibility of more dissolved phosphorus in the solid wastes can be produced [57,64]. Additionally, the use of phytase in fish feed is still unproven if we compare it with the poultry and swine feed. Another important consideration regarding phytase activity is that it greatly relies on the gut pH and the water temperature.

#### 1.2.2. Nutritional Disadvantages of Terrestrial Crop Oil

Long-chain polyunsaturated fatty acids (LCPUFA) such as docosahexaenoic acid (DHA) and eicosapentaenoic acid (EPA) are the major limiting fatty acids acid in terrestrial crop oils. These are the essential omega-3 fatty acids for salmonids, and the requirement is around 0.5–1.0% of the diet [65]. Some studies have shown that the full replacement of fish oil with alternative oil sources can be possible without compromising fish performance, survival, and feed efficiency over the entire production cycle [66,67,68,69]. In the fishmeal-based diets, the essential fatty acids requirement can be met from the fishmeal; thus, the complete substitution of fish oil can be possible using vegetable oils, without growth retardation or any negative health effects [70,71,72]. Salmonids have a highly efficient protein-sparing capability and lipid utilization efficacy. This specifically favors replacing fish oil with vegetable oil especially in fishmeal-based diets for salmonid aquacultures.

#### 1.2.3. Fillet Quality for Human Consumption

The composition of omega-3 and other nutrients in farmed fish species has been altered, and this affects human nutrition, as there are decreases in the mineral concentration of iodine, selenium, vitamin D, and most significantly, omega-3 fatty acids in fish fillet. High percentages of vegetable oils in aquafeeds alter fish muscle fatty acid composition, notably reducing omega-3 such as docosahexaenoic acid (DHA) and eicosapentaenoic acid (EPA), which aids with cardiovascular, cognitive, and neural development and provides other human health benefits [73,74,75]. Partially or totally replacing fish oil with crop oil also unfavorably changes the flesh fatty acid composition in many fish species [68,69,74]. Salmonids fillets are marketed for their human-health-benefitting properties, mainly for their high omega-3 content. This benefit of consuming farmed salmonids has decreased in recent years because current feed formulations contain higher levels of vegetable oils and lower levels of omega-3 DHA and EPA. Over the past decade, the beneficial EPA and DHA content in farmed salmonids fillets has significantly declined, mainly due to the higher inclusion of vegetable oils in aquafeeds (Figure 1) [76,77]. A recent study conducted by the University of Stirling displayed that one portion of Scottish salmon would provide 3.9 g of omega-3 (EPA and DHA); however, this was reduced to 1.9 g of EPA + DHA in 2016, because the use of terrestrial crop protein and vegetable oil increased in salmonid feed [77].

The American Dietetic Association and Dietitians of Canada recommends a daily consumption of 500 mg of n3 LC PUFAs (EPA + DHA), which is the equivalent of two fish servings/week of approximately 100 g of cooked (130 g of raw) fatty fish [78,79]. Several studies have reported that most salmonids held in freshwater potentially have the ability to convert C18: 3n-3 (a-linolenic acid) from vegetable oil into the corresponding C20:5 n-3 (EPA) and C22:6 n-3 (DHA) in vivo via an alternating succession of desaturation and elongation [80,81,82,83,84]. However, the de novo synthesis of any nutrient including DHA can increase metabolic energy expenditure, which might be better used to support somatic growth.

It is also well known that it is easy to formulate a commercial tilapia diet with low fishmeal and no fish oil (with vegetable oil) that will not decrease growth [85,86]. This commercial formulation, however, causes an unbalanced ratio of n3:n6 in the fillet due to a reduced deposition of n3 and increased deposition of n6 [87,88]. Recently, the New York Times highlighted this issue and reported that farmed tilapia, raised largely on a commercial diet containing corn and soy, had a high n6 deposition [89]. This has led medical doctors and nutritionists to doubt the nutritional benefits of eating farmed tilapia. Several studies of current tilapia feeds show elevated levels of C18 2n-6 (linoleic acid) fatty acids from crop oils [90,91], which greatly alters the fatty acid composition of tilapia flesh [86] and can contribute to an imbalanced n-3/n-6 ratio in human consumers [88], causing an increased production of proinflammatory eicosanoids via C20:4n-6 (arachidonic acid). The desired ratio of omega-3 to omega-6 fatty acids in a person’s diet is about 1:1 for optimum health benefits [92,93,94]. Wild tilapia have n3:n6 ratios ranging from 2.6:1 to 6.4:1 and contain more n3 LC PUFAs than farmed tilapia because wild fish eat natural foods rich in n3 LC PUFA [95,96]. It is biologically possible for farmed tilapia to meet the goal of the complete replacement of fish oil using *Schizochytrium*-dried whole-cells, which leads to an omega-3:omega-6 ratio of 1.4:1 in tilapia fillets, which is supportive of a more favorable ratio (1:1) for human consumers [97].

Although terrestrial crop protein and vegetable oil cause significant environmental harm and have nutritional disadvantages, these are the largest alternative feed ingredients mainly due to their wide availability and the lower price of these commodities compared to FMFO. Currently, aquafeed producers are seeking more sustainable and commercially scalable substitutes for fishmeal and fish oil due to rising prices, the price volatility of these commodities, food security concerns, and environmental harm.

## 2. Novel Aquafeed Ingredients

### 2.1. Microalgae Protein and Oil

The industrial-scale production of microalgae has gained momentum for their use in aquaculture feeds [98,99,100]. Marine microalgae particularly have potential to replace fishmeal and fish oil in salmonids and other finfish feeds because of their high levels of fatty acid and protein content. Marine microalgae, *Nannochloropsis oculata*, *Isochrysis* sp., and *Schizochytrium* sp. showed promise in aquafeed because they are rich in EPA, DHA, protein, key amino acids (methionine and lysine), lipids, and are good sources of minerals. A recent study showed that *Isochrysis* sp. is a highly digestible protein, amino acid, lipid, and fatty acids source for rainbow trout (Table 1). This species could be a good candidate for fishmeal and fish oil replacement in rainbow trout diets and can be used as a health-promoting omega-3, DHA supplement in diets [100]. Research showed that lipids extracted from *Desmodesmus* sp. could be used (20%) in the salmon feeds without any negative effect on growth and fillet composition [101,102]. A *Spirulina* algal meal could be also incorporated in 10% of the rainbow trout feed without any adverse effect on fish performance [102].

Defatted *N. oculata* coproducts (leftover biomass oil extraction) contain approximately 20% to 45% crude protein, with good amino acid profiles. Including defatted *N. oculata* as a protein source into diets up to 33% for tilapia [103] and up to 10% for Atlantic salmon [104] did not affect their performance or health status. Another unique benefit of the defatted microalgae is their function in serving not only as an excellent protein source but also as a source of polyunsaturated fatty acids (PUFAs) to enrich n-3 fatty acids. Defatted *N*. *oculata* is more nutrient-dense compared to the whole cells. The digestibility of lysine (often deficient in terrestrial crop protein) was higher [103], and the EPA was also highly digestible, making it a good candidate for EPA supplementation in tilapia feed [103]. However, the defatted *N. oculata* caused a significantly lower digestibility of protein and methionine than the whole cells in tilapia. The decreased digestibility and growth at a higher inclusion rate of *N*. *oculata* indicated higher levels of antinutrients in the defatted *N. oculata* coproducts than the whole cells.

**Table 1 microorganisms-11-00439-t001:** Summary of microalgae (*Schizochytrium* sp. and *Isochrysis* sp.) whole cells used as an oil source in research.

Microalgae	Fish Species	FCR	Inclusion % of Diet	Reference
*Schizochytrium* sp.	Atlantic salmon	1.40 and 1.42	5.5 and 11	[105]
*Schizochytrium* sp.	Tilapia	1.0, 1.0, 0.9 and 0.9	4, 8, 12.5, and 16.1	[97]
*Schizochytrium* sp.	Tilapia	1.57, 1.60 and 1.4	6.20	[29]
*Schizochytrium* sp.	Rainbow trout	1.0 and 0.98	9 and 6.5	[41]
*Schizochytrium* sp.	Pacific white shrimp	Not reported	0.6, 1.2, 1.8, 2.3 and 3.5	[106]
*Schizochytrium* sp.	Atlantic salmon	0.72 and 0.73	2.5 and 5.0	[107]
*Schizochytrium* sp.	Rainbow trout	0.98	2.50	[100]
*Isochrysis* sp	Rainbow trout	1.0	2.40	[100]
*Isochrysis* sp	European sea bass	1.69 and 1.76	7 and 14	[108]

Predominantly heterotrophic DHA-rich *Schizochytrium* sp. have been investigated for oil and have been found to be a highly digestible source of nutrients for salmonids, shrimp, tilapia, and other species and a potential substitute for fish oil in aquafeed [1]. A recent study showed a significantly improved gain in weight, feed conversion ratio, and protein efficiency ratio when fish oil was fully replaced using *Schizochytrium* sp. biomass in a tilapia diet [97]. It has been reported to develop a high-performing fish-free feed (full substitution of FM and FO) for tilapia by combining two marine microalgae (one is an *N. oculata* coproduct and another is *Schizochytrium* sp. biomass [97].

A recent study showed promise in developing a fish-free feed for rainbow trout by combining marine microalgae *Nannochloropsis* sp., *Isochrysis* sp., and *Schizochytrium* sp.; the detected fillet DHA levels were similar (Figure 2), although the fish-free feed exhibited a minor, but significant, lower growth (due to lower feed intake) than the fish fed a conventional feed, and this could be improved by adding feeding stimulants [100]. *Schizochytrium* sp. still holds promise as an alternative to fish oil because the production of heterotrophic microalgae is much easier and quicker (a few days) than phototrophic microalgal production via fermentation, which utilizes carbon sources for energy (e.g., corn, sugar cane byproduct). In addition, the cultivation conditions of microalgae, for example, light, temperature, and nutrient source, can also impact the phototrophically grown microalgal composition, including the fatty acid composition. It ultimately affects the consistent quality of the final product. There has been extensive research on the capabilities of heterotrophic algae oils and their success in commercial trials. A number of aquafeed companies have recently begun commercially producing DHA-rich oil from *Schizochytrium* sp. for salmon feeds [109]. Many agribusiness and animal nutrition companies started producing heterotrophic *Schizochytrium* sp. as fish oil substitutes (Corbion; BioMar; Archer Daniels Midland, ADM; and Veramaris). However, extracting oil from a microalgal biomass is a cost-prohibitive process compared to extracting fish oil; for example, pure microalgal oil costs twice as much as fish oil in the market. It is important to see the more commercial scale production, which will in turn reduce the cost of ingredient production. Moreover, performing life cycle environmental analyses of microalgae when producing aquafeed feed is also important to detect the effects on global warming potential, other environmental footprints, and land and freshwater use. Our recent article on the life cycle analysis of *Schizochytrium* sp. in aquafeeds showed that *Schizochytrium* sp. could be a better sustainable fish oil substitute for aquaculture feed due to its lower biotic resource depletion and global warming potential [14]. This outcome helps strengthen sustainable environmental stewardship of aquaculture feeds with *Schizochytrium* sp.

### 2.2. Yeast, Fungi, and Bacteria

*Saccharomyces cerevisiae*, various *Aspergillus*, and *Fusarium venenatum* are widely known, but other strains such as Candida utilis, *Candida*, *Hansenula*, *Pichia*, *Torulopsis*, and *Kluyveromyces marxianus* are also stimulating interest as protein ingredients for aquaculture feed [37,110,111]. Several yeast meals, for example, *S. cerevisiae*, *Candida utilis*, and *K. marxianus* have been used in salmon feeds [110,112]. *Candida utilis* and *K. marxianus* are good sources of protein and could substitute up to 40% of the fishmeal without compromising growth [110]. However, *S. cerevisiae* was reported as a poor protein ingredient for fish feed.

Some bacterial strains can play an important role in producing very high crude protein contents (approximately 60 to 82% of dry cell weight) and essential amino acid levels, along with vitamins, phospholipids, and other functional compounds [113,114]. For example, several methanotrophically grown bacterial proteins have been investigated in Atlantic salmon feeds. A recent study revealed that salmon fed a 36% bacterial protein meal had a better growth rate and feed efficiency ratio than when fed the reference diet [115]. KnipBio Meal (*Methylobacterium extorquens*) could replace 55% of fishmeal in salmon diets with no adverse side effects upon growth [116]. Bacterial protein also showed good promise in shrimp. KnipBio Meal could replace 100% of fishmeal in shrimp diets [116]. Combining two purple nonsulfur bacteria at 1% of a diet resulted in modest growth improvements [117]. *Corynebacterium ammoniagenes* could replace 10–20% fishmeal in shrimp [118]. Combining bacteria and microalgae (Novacq™, CSIRO Canberra, Australia) has been tested on black tiger shrimp and prawn [119,120]. There is great potential to produce bacterial protein, but more organisms need to be identified for commercial scale production. It is interesting to note that the bacteria can be cultured on agricultural wastes (rice straw, rice hulls, manure, and starchy residue), as the substrates attain a high level of protein [121]. Additionally, bacterial proteins contain other nutrients including lipids, vitamins, and minerals [37,122]. Although bacterial proteins are very attractive for future aquafeed, the challenges associated with processing, the economy of scale, and their adoption globally need to be addressed.

### 2.3. Genetically Modified Oil

Western consumers consumed GM foods on a regular basis for a decade, but the human health effects of eating GM foods are not well understood. Several types of currently available vegetables and processed foods in the market are nonlabeled GMOs. Recently, Cargill deregulated genetically modified transgenic foods (canola oil/camelina oil/yeast). The DHA and EPA content in transgenic oilseed camelina can avoid the production of undesirable fatty acids [123]. GM rapeseed oil and yeast displayed a high retention of EPA and DHA in Atlantic salmon fillets [124]. However, to obtain public confidence and satisfaction, it is critical that future researchers know the long-term effects of the GM ingredients in fillets made from aquaculture-feed-fed farmed fish on human health. Additionally, fish fed GM ingredients should be evaluated to determine the sensory properties of fillet due to potential biochemical changes that could affect sensory qualities. Several commercial industries (for example, Aquaterra/Nuseed and BASF/Cargill) are currently using GM yeast and oilseed crops, which are rich in EPA or DHA or a combination of both. The process involves selecting and cloning the genes involved in EPA and DHA biosynthesis from marine algae and then inserting them into terrestrial crops. However, the public perceives any GM product as still being poor to the users. Additionally, regulatory hurdles still exist, which makes it illegal to grow GM plants in some countries.

### 2.4. Insect Meal

The aquafeed industry is actively seeking a diverse array of alternative feed ingredients for fishmeal, including insect meal. Insects could provide a sustainable source of protein for aquacultures using food waste. As of now, the following species are the most studied for producing protein meals, and they account for the majority of the literature: black soldier fly (*Hermetia illucens*, L.; BSF), common housefly (*Musca domestica*), yellow mealworm (*Tenebrio molitor*), lesser mealworm (*Alphitobious diaperinus*), house cricket (*Acheta domesticus*), banded cricket (*Gryllodes sigillatus*), and field cricket (*Gryllus assimilis*). Among these species, the BSF is considered the most attractive insect meal for aquafeed for salmonids, i.e., rainbow trout (*Onchorhynchus mykiss*) and Atlantic salmon (*Salmo salar*) [125,126,127,128,129,130,131]. Several studies have shown promise with the use of insect meals in aquafeeds, and the inclusion of insect meals is still a recent development, with many uncertainties existing that could influence whether the aquafeed industry adopts insect meals on a large scale. More research is needed to formulate feeds for specific aquaculture species. The authors of [130] for example, demonstrated the sensitivity of greenhouse gas emission results to direct emissions from the insects during growth, for which very limited data were available. Moreover, the consistent quality and quantity of supply is important for commercial feed mills to fine tune feed formulations, as the raw materials have to be the same every batch. For example, if we feed our insects different waste streams as substrates, their nutritional properties will be altered and inconsistent. Additionally, cannibalism behavior may prevent the insects from being raised uniformly, as the smaller larvae are eaten by the bigger ones. We need more research on unconventional aquaculture feed ingredients to solve these problems. Additionally, data on the antinutrient compositions (for example, chitin, protease inhibitors, oxalate, tannin, etc.), palatability, and digestibility of insect meals are lacking. Brewery and distillery wastes as byproducts are currently largely used in poultry and salmon feeds. It seems like it is not worthwhile to feed the nutrient-rich wastes to insect larvae when we could feed them directly to livestock. The use of wastes to grow insects is important, as an insect grower could target certain wastes that have no other use, although more research is needed on aspects such as the biosecurity issue. Insect larvae could be raised on a large volume of industrial manure (industrial farm), but farmers should be careful about antibiotics and other contaminants.

### 2.5. Fish Processing Byproduct

The processing of fish for human food generates byproducts such as heads, viscera, frames, skins, and others, such as tails, fins, scales, mince, blood, etc. These byproducts are actually good sources of protein and oil, from which fishmeal and fish oil can be yielded. Fish-processing byproducts are still not fully utilized. Approximately 25–35% of fishmeal comes from the byproducts of fish processing [132]. Approximately 10% of byproducts are generated by aquacultures, 19% of byproducts are generated by capture fisheries, and 71% of byproducts are generated by whole capture fisheries.

Currently, the byproducts of herring and tuna fisheries provide about 5 million metric tons, which accounts for around 25 percent of global fishmeal production. For example, the processing of Atlantic salmon (*Salmo salar*) fillet yields about 60% of byproducts that contain high levels of omega-3 DHA and EPA. Approximately 11.7 million tons of additional byproducts are generated in fish processing plants that are not certified for the production of fishmeal and fish oil [132]. The largest volume of fish processing waste is produced in North America and Oceania. Fish wastes are mainly generated in the global south, mainly due to a lack of high-quality ice, cold storages, and refrigerated transportation [3]. Capturing fish processing byproducts is often not economically viable due to logistic and technical constraints.

## 3. Key Strategies to Measure Nutritional and Environmental Sustainability

### 3.1. Ingredients Digestibility

The digestibility of aquafeed ingredients is key information for formulating economically viable and environmentally responsible feeds, but limited digestibility data are publicly available for alternative ingredients [100]. This lack of information often leads aquafeed manufacturers to extrapolate the nutritive value of ingredients from their chemical composition. Because a simple biochemical analysis and the presence of protein and amino acids in the ingredients does not ensure the level of digestible proteins and amino acids required for particular fish species, it is important to conduct digestibility experiments to identify the digestibility of ingredients and formulate the diet based on the digestible nutrient content, which can reduce feed costs, nutrient pollution (including phosphorus and nitrogen eutrophication emissions), and improve the feed conversion ratio of aquafeeds. The main reason for the improved feed conversion ratio (FCR) achieved today is due to the significantly increased digestible nutrients in and energy density of the feeds. In addition to the digestibility data, the aquafeed industry will be benefitted by using alternative ingredients in feed formulations if they have specific information on parameters such as the nutrients and antinutrient composition (for example, pectin, lectin, chitin, protease inhibitor, oxalate, tannin, etc.).

### 3.2. Feed Conversion Ratio (FCR)

The development of sustainable feed using alternative ingredients should be informed by the efficiency of feed with a lower FCR. A common measure of efficiency is the feed conversion ratio (FCR), calculated as the ratio of feed intake to weight gain. It reflects the environmental performance of the aquaculture because it provides an indication of the phosphorus and nitrogen waste outputs in the aquatic environment with potential negative consequences of eutrophication, greenhouse gas emissions, loss of biodiversity, and other ecosystem services. Additionally, the contribution of aquacultures to greenhouse gas emission is strongly related to the FCR and the origin of the feed components. Although, the FCRs of aquacultures have fallen over the past several decades from around 3 to around 1.35, largely because of better feed formulations, feed manufacturing methods, and on-farm feed management. Specifically, salmonids farming improved the FCR from about 2–2.25 to approximately 0.9–1.2 since 1970 due to better feed formulation using highly digestible feed ingredients, technological developments, and on-site feed management. Now it is time to make more sustainable feed formulations for other species such as carps, catfishes, tilapia, and marine shrimps. The feed conversion ratio reflects the fact that fish are the most efficient species among all commercially farm-raised animals. For example, farmed fish can convert approximately 1.35 kg of feed (dry) into 1 kg of flesh (wet); in contrast, the feed conversion of poultry is 1.8, pork is 3.5, and beef is 6.25 [9,11,37].

### 3.3. Life Cycle Assessment (LCA) for Ecological Impact Measures

The environmental impacts of aquafeed can be evaluated with a LCA, which is an important tool to measure the environmental impacts of food systems [53,133,134]. The development of sustainable feed using alternative ingredients should be informed by environmental impact categories including resource use (e.g., land, water, fertilizer), global warming emissions, eutrophication emissions, biodiversity loss, and negative externalities including ocean acidification [53,135]. Very limited data on life cycle environmental impacts and emissions from novel ingredients are available. The categories of the environmental footprint, determined with an LCA, should be included for novel ingredient evaluation processes [136,137]. Additionally, there is no food production system today that is truly sustainable in terms of energy use and biodiversity loss. However, this information is essential to evaluate the performance of novel ingredients and compare them with FMFO production. The alternative aquafeed industry is in its infant stages, and the broader environmental burdens of producing novel ingredients have not yet been established. A more holistic view of sustainability is therefore needed. Thus, mass producing new ingredients could depend on fossil fuels, and creating high-quality emerging protein and oil requires us to learn about the life-cycle effects of these ingredient substitutes on FMFO in diets. It is important to conduct more research on the LCAs of currently available unconventional feedstuff to better understand the more sustainable future ingredients for aquacultures.

## 4. Novel Technology for Improving Ingredients Quality

Improving the overall feed efficiency/feed conversion ratio and digestibility of protein ingredients will help the aquafeed industry meet the global demand for these limited ingredients. There are several factors that are responsible for enhancing feed efficiency and formulating least-cost diets: (i) the quality of the protein ingredients, which includes the protein concentration, amino acid profile, and antinutrient factors; (ii) the ability to meet protein and essential amino acid requirements for a given fish species at life stages because the relative ontogenetic stage of the fish can significantly affect the protein or essential amino acid levels required in their diet; and (iii) the ability of fish to digest and retain the ingested feed protein. New technologies have been developing to process the ingredients that can improve the feed efficiency/feed conversion ratio, digestibility, and protein efficiency. For example, microalgae are still in the infant stages of being included in aquaculture diets, and the further processing of these ingredients may enhance both the protein and energy digestibility. The nutrient digestibility of microalgae ingredients can be an issue because of the rigidity of their cell wall [97,138]. To improve the digestibility of microalgae, the cell wall can be ruptured via several methods [138,139]: enzymatic (cellulases, xylanase, glucanase), chemical (organic solvents or acids), and physical and mechanical (bead milling, extrusion, grinding, autoclaving, roasting/toasting, high-pressure homogenization, or microfluidics). Physical processing is effective at partially or completely inactivating the antinutritional factors. Physical and mechanical methods are generally preferred [140,141], as enzymatic and chemical methods can impact intracellular nutrients. Trypsin inhibitors and lectins in soybeans could be inactivated by heating or cooking [142]. A recent study showed that the whole-cells of Chlorella and the ruptured cells had the same nutrient content, but the ruptured cells had improved digestibility over the whole cells of Chlorella for essential amino acids, lipids, and carbohydrates [138].

### 4.1. Extrusion Processing

The novel extrusion technology offers the advantage of making use of a wider variety of ingredients. For example, twin screw extruder technology can be used to process, stabilize, and incorporate ingredients. The extruder has various capacities, along with drying, grinding, and other related equipment to conduct pilot-scale manufacturing tasks. Extrusion processing exposes ingredients to high temperatures and pressures over a short period of time, resulting in cooking and pasteurization and thus eliminating any antinutrients in the ingredients and increasing the feed intake, nutrient digestibility, and therefore the growth performance of the fish [63,104,143,144,145]. Additionally, extruded feed allows higher lipid levels to be included in the diets, and the gelatinization of starch stimulates the increase in the protein and energy digestibility of feeds. Extrusion also helps to produce more durable feed, with less fines and the ability to vary the physical characteristics of the pellet (buoyancy or sinking).

### 4.2. Use of Exogenous Enzymes in Aquafeed

Protease enzymes may stimulate endogenous peptidases by improving protein digestibility and hydrolyzing proteinaceous antinutrients such as lectins, trypsin inhibitors, antigenic proteins, and antinutritional allergenic proteins such as glycinin, β-conglycinin, and kafrin [146]. High-quality proteases or cocktails can be developed for specific proteins (e.g., the keratin-rich poultry feathers) based on the digestive conditions (e.g., stomach or intestinal pH), which could help enhance dietary protein digestion and utilization. The digestive tracts of monogastric animals, such as rainbow trout, lack any appreciable nonstarch polysaccharide enzyme activity [56]. Thus, treating NSP enzymes (xylanases, glucanases, cellulases) to ingredients could enhance the digestibility and utilization of nutrients supplied by alternative ingredients. Supplementation with these enzymes has shown increased nutrient digestibility, nutrient utilization, fish growth, reduced nutrient wastes, and antinutritional factors in terrestrial plant ingredients for fish including rainbow trout [147,148,149].

### 4.3. Use of Additives in Aquafeed to Improve Palatability

The supplementation of taurine with alternative ingredients would enhance the palatability of low fishmeal or fish-free feeds and improve digestion and lipid adsorption. Taurine is a neutral beta-amino acid derived from the metabolism of sulfur containing amino acids. In the past, taurine was not considered an essential nutrient for fish, but recent research showed that it plays a key role in aquaculture nutrition to improve growth feed intake and feed utilization [144,150,151,152]. Usually, fish-based diets are rich in taurine; therefore, salmonids or other carnivorous fish may require exogenous taurine in fish-free feed to maintain their physiological functions and increase their feed intake. The addition of lecithin to aquafeed containing low levels of FM has also been shown to improve feed intake and growth rates in trout, salmon, and flounder [153,154,155].

## 5. Challenges with Adopting and Opportunities to Adopt the New Aquafeeds

The typical rate (share of individual dietary components in feeds) of soy and corn combined is about 40–60% in fish feeds [97,100,103]. Replacing terrestrial crops with the other current alternatives in fish feed is likely to be too costly for broader adoption by aquafeed manufacturers and aquaculture producers. The aquafeed industry will use alternative ingredients only if it is at a cost competitive with soy and corn and at a constant quantity of supply [103,138]). Thus, researchers and aquaculture industry stakeholders need a collaborative effort and need to pursue stepwise research to identify a prime alternative that will be productive and profitable while minimizing negative environmental, social, and economic impacts. Towards this end, scientists should be pursuing stepwise research to find a sustainable solution, whether it is a novel ingredient or a combination of these ingredients or a coproduct or byproduct that can partially or fully replace crop proteins in aquafeeds, and to determine how this combination affects fish growth, flesh quality, and economic and environmental performances.

An evaluation of the sustainability of new ingredients is gaining attention. Several approaches were applied to define sustainability, but a life cycle assessment is perhaps the best approach. Our lab is now developing an open-access decision-support tool that integrates a technoeconomic and life cycle analysis and a nutritional analysis of diverse alternative ingredients, from insect meals to microalgae for fish-free aquaculture feeds.

However, we should acknowledge that the major challenge is generating consistency with the supply of microalgal ingredients to produce large quantities on an industrial scale. Another big challenge is the competitive price of unconventional ingredients with conventional ingredients. Price is the major constraint that will dictate the future course of the aquafeed sector and industry. Thus, scaling up is a real block for these novel ingredients. The big challenge is the competitive price of novel ingredients with conventional ingredients. Aquaculture feed manufacturers are willing to pay a similar price per ton for soy protein concentrate as fishmeal. As traditional fish feed ingredients make up a significantly large portion of aquafeed today, the focus has just begun to include a minor percentage of new raw ingredients because there is still not any solid guarantee of consistent supplies and because of the competitive costs. Researchers and microalgae R&D have to find ways to lower the costs of producing and processing novel ingredients. For precise feed formulations, the raw ingredients have to be the same in every batch. For example, if an insect is being produced via different waste streams, then their nutritional compositions will be inconsistent. Relatively small production volumes of novel ingredients such as microalgae, bacteria, yeast, and insect meal still supply a limited market contribution compared with traditional feed ingredients. For example, although replacing fish oil with microalgae showed promise, producing microalgae and extracting oil requires access to advanced technologies, expertise, and money. Moreover, the use of these novel ingredients is not cost effective for small-scale aquaculture producers and notably for developing aquaculture nations. Thus, at this moment, adopting novel ingredients is still challenging and may not be equitable to all producers across the world. There are multiple regulatory, economic, and environmental challenges that accompany switching from conventional aquafeeds to ones based on innovative ingredients, especially for smaller producers and producers in developing countries [33]. Despite the above limitations, if the stakeholders agreed to adopt novel ingredients for aquafeeds where it was feasible, this would be an important step towards reducing pressure on wild-caught fish in the aquaculture diet, which could secure the future sustainability of the aquaculture sector. Exploring economic incentives, investigating marketing opportunities, and promoting the ecocertification of sustainable ingredients is also needed. Consumers’ preferences for and awareness of sustainability is growing, and there is a positive public attitude towards novel ingredients, as expressed by a willingness to pay a sustainable premium for its products, which can further enhance the profitability of adopting novel ingredients.

## 6. Conclusions and Future Steps

Regarding progress towards the development of novel ingredients, it is imperative to determine the nutritional evaluations of each ingredient, which includes the determination of nutritional value, palatability and feed intake, and digestibility, and requires researchers to conduct nutritional feeding experiments in the lab and feeding/growth trials at commercial farms [156]. Some ingredients may require additional processing to improve digestibility, the FCR, and reduce the antinutritional factors. The various processing could also increase the opportunities to produce lower-cost protein-rich ingredients and might reduce the overall costs of production. In the final stage of the nutritional evaluation of alternative ingredients, a comprehensive economic analysis is also important to inform decisions about the inclusion of unconventional ingredients into diets that are cost competitive with conventional ingredients. Additionally, some recent assessments showed that it might be possible to produce yields of novel ingredients such as microalgae (per unit area) several times higher than that of terrestrial crops (soy protein) using the same land footprint using atmospheric carbon and renewable energy [157]. As mentioned, the sustainable production approach clearly provides an indication of a new emerging priority to conduct a life-cycle assessment analysis of new ingredients [136,137]. Thus, to fully develop sustainable alternative ingredients, scholars must use systems-based approaches that integrate knowledge and expertise across multiple disciplines including fish nutritionists and aquaculture scientists to conduct on-farm feeding trials with farmers, environmental scientists, technoeconomic and life-cycle practitioners, extension scientists, and economists.

## Figures and Tables

**Figure 1 microorganisms-11-00439-f001:**
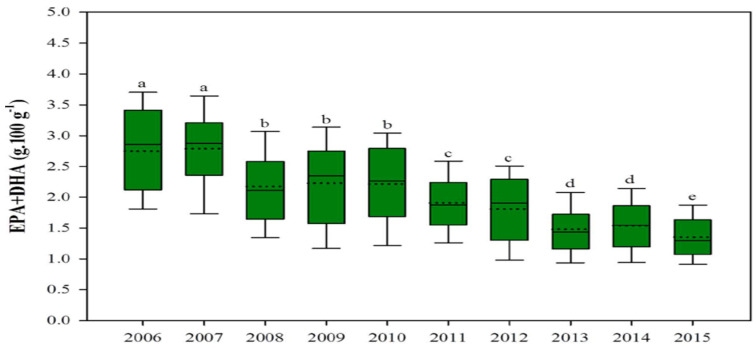
Contents of EPA and DHA (omega-3) in farmed salmon have fallen significantly (source: adapted from [77] CC 4.0). For a, b, c, d in the figure caption meaning both letters appearing together means no difference.

**Figure 2 microorganisms-11-00439-f002:**
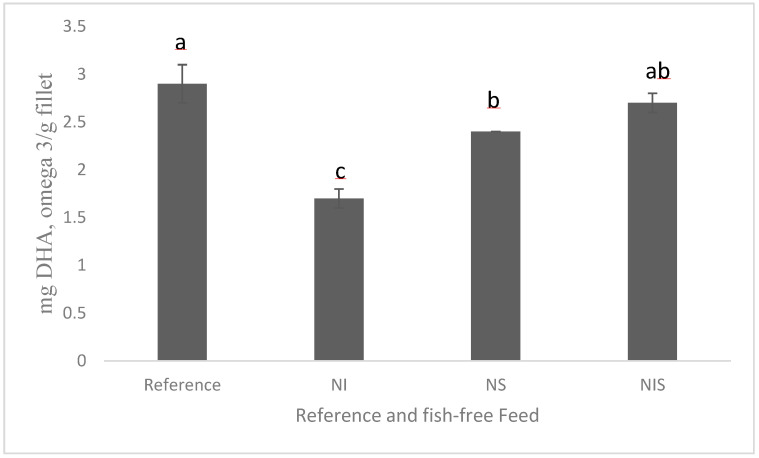
Amounts of DHA in the fillet of rainbow trout fed experimental diets for 84 days (adapted from [100] CC 4.0). NI = *Nannochloropsis* sp. + *Isochrysis* sp.; NS = *Nannochloropsis* sp. + *Schizochytrium* sp.; NIS = *Nannochloropsis* sp. + *Isochrysis* sp. + *Schizochytrium* sp. For a, b, c, d in the figure caption meaning both letters appearing together means no difference.

## Data Availability

All data generated by this study are presented in the tables.

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
