# Peer review of "Microorganisms in Fish Feeds, Technological Innovations, and Key Strategies for Sustainable Aquaculture"

_microorganisms, 2023, doi:10.3390/microorganisms11020439_

Round 1

Reviewer 1 Report

In my point of view, the manuscript is judged to be of potential interest to the readers and the scientific community as well. While the overall concept of the paper is reasonably satisfactory, it falls within the scope of the journal. However, it has some corrections. The manuscript still has a number of flaws, which need to be addressed before it is formally accepted for publication.   Overall, this manuscript may offer important contributions to the literature, but needs some major corrections for possible publication in this journal.

1. Line 42 Please confirm the quote format.

2. Line 58-60 feed efficiency ratio(FER)should be improved and fcr should be decreased.

3. Line 61-64 Confirm explanation of protein efficiency ratio.

4. Line 110-112 The sentence is repeated.

5. Line 156-167 The author discusses the nutritional deficiencies of crop oils, but concludes by saying that vegetable oils are a good substitute for fish oil. Please make changes.

6. Line 414 FCR should be decreased.

7. Line 453-465 The authors mention the use of enzyme and physicochemical methods to solve the problem of microalgal digestibility. This method can also be used for plant materials. Since both need to be treated for use, why do the authors emphasize microalgae?

Author Response

Thank you so much to reviewer 1 for giving intellectual input and tremendous efforts to review the manuscript carefully. We have carefully addressed all the points that have been raised by the reviewers. I am submitting the revised version of our manuscript (track-changed) after thorough and careful revisions in response to the reviewers’ helpful suggestions. Thus, please find attached our responses to those reviews.

We have revised the manuscript to fix the main points raised by the reviewers. We also revised all required passages to address the reviewers’ more major and minor comments.

Thank you in advance for your time and effort in considering this work. Responses are highlighted in red type below:

  1. Line 42 Please confirm the quote format: Addressed as suggested in revised version showing in track change.
  2. Line 58-60 feed efficiency ratio(FER)should be improved and fcr should be decreased. Addressed
  3. Line 61-64 Confirm explanation of protein efficiency ratio. Explained in parenthesis
  4. Line 110-112 The sentence is repeated. Thank you for the good catch. Deleted the repeated sentences.
  5. Line 156-167 The author discusses the nutritional deficiencies of crop oils, but concludes by saying that vegetable oils are a good substitute for fish oil. Please make changes. Good point. Clarified and changed as suggested
  6. Line 414 FCR should be decreased. Yes, it obvious when FCR improved, it indicates FCR should be lower’
  7. Line 453-465 The authors mention the use of enzyme and physicochemical methods to solve the problem of microalgal digestibility. This method can also be used for plant materials. Since both need to be treated for use, why do the authors emphasize microalgae? Microalgae is more sustainable than plant materials. As outlined in the manuscript there are several nutritional disadvantages and in addition to that the overreliance of aquafeed on terrestrial crops can drive massive use of farmlands to grow crops for fish feed (including current livestock feeds), instead of for direct human consumption (food security in question). Reformulating the composition of aquafeed using microalgae, thus, is a key leverage point for reforming aquaculture so that it helps conserve rather than damage natural ecosystems and human and fish health.

Reviewer 2 Report

Specific comments:

Line 65: „of the total cost“ is mentioned twice, remove one.

Line 77-99, 280-291: Font of the chapter should be equalized with the rest of the text.

Line 84: Give full name of the acronim FMFO on its first metioning.

Line 85: Is it 24% of total FM and 50% of total FO?

Line 88: Is it 2 to 4 % of total FMFO?

Line 108: change to „typically“

Line 186: The reference for the statement on higher inclusion of vegetable oils in aquafeed is not correct because Figure 1 is not showing that.

Lines 227-228: It should be: fatty acids and proteins

Line 240: Reference Sarker et al. 2018 should be written in the same form as all other references.

Line 248: methionine were significantly what?

Line 251: Table 1 has no reference in the text. References in the table have different format from the rest of the manuscript, but the format should be the same.

Line 260: Figure 3. has no reference in the text. There's no reference in the caption of the figure.

Line 293-299: Only latin names of microorganisms should be written in italic font.

Line 301: Open paranthesis shoud be in front of word „approximately“.

Line 374: Capture fisheries is mentioned twice.

Line 861 and line 868: The same reference is presented twice, remove one.

Line 888: The reference is incomplete. Journal name, volume and pages are missing.

Author Response

Thanks very much to reviewer 2 for giving intellectual input and tremendous efforts to review the manuscript carefully. We have carefully addressed all the points that have been raised by the reviewers. I am submitting the revised version of our manuscript (track-changed) after thorough and careful revisions in response to the reviewers’ helpful suggestions. Thus, please find attached our responses to those reviews.

We have revised the manuscript to fix the main points raised by the reviewers. We also revised all required passages to address the reviewers’ more major and minor comments.

Thank you in advance for your time and effort in considering this work. Responses are highlighted in red type below:

Specific comments:

Line 65: „of the total cost“ is mentioned twice, remove one. Removed as suggested in revised version showing in track change. 

Line 77-99, 280-291: Font of the chapter should be equalized with the rest of the text. Equalized

Line 84: Give full name of the acronim FMFO on its first metioning. Inserted the acronym in the revised version

Line 85: Is it 24% of total FM and 50% of total FO? Yes, inserted total

Line 88: Is it 2 to 4 % of total FMFO? Percent of diet not total

Line 108: change to „typically“ Corrected

Line 186: The reference for the statement on higher inclusion of vegetable oils in aquafeed is not correct because Figure 1 is not showing that. Great point, corrected and it should be Figure 2

Lines 227-228: It should be: fatty acids and proteins added “and”

Line 240: Reference Sarker et al. 2018 should be written in the same form as all other references. Changed the format

Line 248: methionine were significantly what? Reframed the sentence

Line 251: Table 1 has no reference in the text. References in the table have different format from the rest of the manuscript, but the format should be the same. Changed the references format.

Line 260: Figure 3. has no reference in the text. There's no reference in the caption of the figure. Addressed

Line 293-299: Only latin names of microorganisms should be written in italic font. Revised

Line 301: Open paranthesis shoud be in front of word „approximately“. Done

Line 374: Capture fisheries is mentioned twice. Revised

Line 861 and line 868: The same reference is presented twice, remove one. Removed

Line 888: The reference is incomplete. Journal name, volume and pages are missing. Included

Round 2

Reviewer 1 Report

  • Good modifications have been made to the standard of acceptance.